# Ultraviolet Light Treatment of Titanium Microfiber Scaffolds Enhances Osteoblast Recruitment and Osteoconductivity in a Vertical Bone Augmentation Model: 3D UV Photofunctionalization

**DOI:** 10.3390/cells12010019

**Published:** 2022-12-21

**Authors:** Hiroaki Kitajima, Makoto Hirota, Keiji Komatsu, Hitoshi Isono, Takanori Matsuura, Kenji Mitsudo, Takahiro Ogawa

**Affiliations:** 1Division of Regenerative and Reconstructive Sciences and Weintraub Center for Reconstructive Biotechnology, UCLA School of Dentistry, Los Angeles, CA 90095-1668, USA; 2Department of Oral and Maxillofacial Surgery, Yokohama City University Graduate School of Medicine, 3-9 Fuku-ura, Kanazawa-ku, Yokohama 236-0004, Kanagawa, Japan; 3Department of Oral and Maxillofacial Surgery/Orthodontics, Yokohama City University Medical Center, 4-57 Urafune-cho, Minami-ku, Yokohama 236-0004, Kanagawa, Japan

**Keywords:** UV photofunctionalization, osteoblasts, osseointegration, bone augmentation, implant

## Abstract

Vertical bone augmentation to create host bone prior to implant placement is one of the most challenging regenerative procedures. The objective of this study is to evaluate the capacity of a UV-photofunctionalized titanium microfiber scaffold to recruit osteoblasts, generate intra-scaffold bone, and integrate with host bone in a vertical augmentation model with unidirectional, limited blood supply. Scaffolds were fabricated by molding and sintering grade 1 commercially pure titanium microfibers (20 μm diameter) and treated with UVC light (200–280 nm wavelength) emitted from a low-pressure mercury lamp for 20 min immediately before experiments. The scaffolds had an even and dense fiber network with 87% porosity and 20–50 mm inter-fiber distance. Surface carbon reduced from 30% on untreated scaffold to 10% after UV treatment, which corresponded to hydro-repellent to superhydrophilic conversion. Vertical infiltration testing revealed that UV-treated scaffolds absorbed 4-, 14-, and 15-times more blood, water, and glycerol than untreated scaffolds, respectively. In vitro, four-times more osteoblasts attached to UV-treated scaffolds than untreated scaffolds three hours after seeding. On day 2, there were 70% more osteoblasts on UV-treated scaffolds. Fluorescent microscopy visualized confluent osteoblasts on UV-treated microfibers two days after seeding but sparse and separated cells on untreated microfibers. Alkaline phosphatase activity and osteocalcin gene expression were significantly greater in osteoblasts grown on UV-treated microfiber scaffolds. In an in vivo model of vertical augmentation on rat femoral cortical bone, the interfacial strength between innate cortical bone and UV-treated microfiber scaffold after two weeks of healing was double that observed between bone and untreated scaffold. Morphological and chemical analysis confirmed seamless integration of the innate cortical and regenerated bone within microfiber networks for UV-treated scaffolds. These results indicate synergy between titanium microfiber scaffolds and UV photofunctionalization to provide a novel and effective strategy for vertical bone augmentation.

## 1. Introduction

Titanium is an osteoconductive material routinely used in dentistry and orthopedics as an implant material in repairs and reconstruction of teeth, joints, and bony sites [1,2,3,4,5,6,7,8,9]. The integration of titanium into bone is known as osseointegration, and its strength depends on the surface texture of the titanium [6,8,10,11,12,13,14,15,16,17]. “Rough” surfaces commonly used in modern implant therapy are characterized by a three-dimensional (3D) micro-morphology of peaks and valleys that improve the capability and strength of osseointegration [4,5,6,13,18,19,20,21,22]. From the mechanical perspective, surface roughening seems to provide a thin layer of micro-scaffolding in which de novo bone tissue juxtaposes two-dimensionally with a titanium surface [10]. Such rough surfaces can achieve rigid bone integration without a tissue gap or intervening soft tissue [1,2,5,12,16,23,24,25,26,27,28,29,30,31]. However, a major challenge to successful implant therapy is overcoming unfavorable host conditions including insufficient width and height of innate bone to receive an implant [32,33,34,35,36,37,38,39,40]. To achieve this, the execution and outcome of vertical bone augmentation require significant improvement [32,41,42,43,44,45,46,47,48,49,50,51].

In addition to increasing interlocking with bone, a biological advantage of microrough titanium surfaces is that they promote the differentiation of adherent osteogenic cells [1,2,3,6,8,12,13,15,26,28,29,52,53]. However, this advantage is limited by decreased proliferation as these cells differentiate [10,14,15,17,54,55,56]. In addition to reduced proliferation, the number of cells migrating and attaching to microrough surfaces is lower than on relatively smooth surfaces [17,21,29]. This is probably less of an issue for the microrough surfaces of bulk titanium devices, such as titanium implants, because there is ubiquitous 360° blood and cellular supply to a large implant area from the bone marrow. When a scaffold made of titanium or other biomaterials is used to augment bone, however, the supply of blood and cells is extremely limited and usually only from one or two directions. It is crucial to optimize the local environment to facilitate blood and cell infiltration into the scaffold.

We hypothesized that a scaffold made of titanium microfibers would provide a structure that enhances bone augmentation [57,58]. Microfibers can create a 3D microrough environment, with fibers acting as peaks and pores as valleys. Indeed, 50 µm-thick titanium fiber scaffolds coated with submicron hydroxyapatite (HA) induced bone healing of a non-critical cranial defect in rats [59,60]. However, HA coating poses a risk of dissociation and inducing an inflammatory and osteoclastogenic microenvironment. In addition, relatively large 50 µm fibers create pore sizes >100 µm, which may be too porous to retain 10 µm-diameter cells within the fiber network. We previously developed a 20 µm titanium fiber scaffold and restored bone continuity of segmental, critical-size mandibular defects in rabbits [61]. However, the osteoconductivity of the scaffold was limited, with <50% bone formation inside the scaffold despite bilateral blood and cellular supply.

Ultraviolet (UV) light treatment of titanium surfaces, referred to as UV photofunctionalization, enhances osteoblast behavior and function and strengthens osseointegration [62,63,64]. The biological effects of photofunctionalization applied to bulk titanium, experimental specimens, or titanium implants have been extensively reported in vitro, in vivo, and clinically [10,32,33,34,35,45,62,63,64,65,66,67,68,69,70,71,72,73,74,75,76,77,78,79,80,81,82,83,84,85,86,87,88,89,90,91,92,93]. The UV light cleans the surfaces of chemically deposited hydrocarbons, rendering the surfaces superhydrophilic [10,66,94,95,96,97,98,99,100], and UV light enhances osteoblast attachment and spreading behavior on titanium surfaces [10,64,65,74,101,102,103,104]. UV treatment is more effective on micro-textured structures [10,64,105], further promoting osteoblast differentiation and faster osseointegration [2,4,6]. Another notable benefit is that UV-treated titanium surfaces suppress oral bacterial attachment and biofilm formation [106,107,108]. In silico simulations have demonstrated that the blood circulation can be improved around UV-treated implants [109].

Here, we hypothesized that UV treatment of titanium microfiber scaffolds would be effective for vertical bone augmentation, including in a model simulating the most difficult environment of an absence of surrounding innate bone and a limited and unidirectional blood and cellular supply. We designed a scaffold of 20 µm titanium fibers to produce a pore size of 20–50 µm to allow cellular infiltration while maximizing retention. We aimed to analyze fluid absorption into titanium microfiber scaffolds with or without UV treatment along with osteoblast behavior and function, followed by in vivo assessment of integration between innate cortical bone and regenerated bone within the scaffold in a challenging vertical augmentation model in rats.

## 2. Materials and Methods

### 2.1. Titanium Microfiber Scaffold Characterization, UV Photofunctionalization, and Liquid Infiltration

Titanium microfiber scaffold (Hi-Lex Corporation, Takarazuka, Japan) with 87% porosity was manufactured by molding and sintering grade 1 commercially pure titanium microfibers (20 μm diameter). The porosity was calculated by the manufacturer based on the volume and weight of a scaffold and the known density of titanium. Scaffolds in disk (20 mm diameter, 1.5 mm thick) and plate (5 × 3 × 1.5 mm) forms were fabricated. The disk scaffolds were used for in vitro culture experiments and surface morphology evaluation, while the plate scaffolds were for liquid infiltration tests and in vivo surgery. The surface morphologies of the titanium microfibers were examined by scanning electron microscopy (SEM) (Nova 230 Nano SEM, FEI, Hillsboro, OR). The disks and larger plates were used for in vitro studies, and the small plates were used in vivo. Before use, all scaffolds were sterilized in an autoclave at 121 °C for 15 min.

Titanium microfiber scaffolds were treated with UVC light (200–280 nm wavelength) emitted from a low-pressure mercury lamp for 20 min using a photo device (TheraBeam Affiny, Ushio Inc, Tokyo, Japan) immediately prior to use. The UV device uses a low-pressure mercury lamp emitting UVC rays. The distance between the light source and the scaffold was approximately 10 mm. Experimental groups to compare were untreated and UV-treated scaffolds.

X-ray photoelectron spectroscopy (XPS) (PHI Quantera SXM, UPVAC-PHI, Inc, Kanagawa, Japan) was performed to determine the atomic distribution of titanium microfibers. Carbon, oxygen, and titanium were evaluated on scaffolds with or without UV treatment.

The wettability of scaffolds with or without UV treatment was evaluated by measuring the contact angle of a 3 µL double-distilled H_2_O (ddH_2_O) droplet in side-view photographs. The ability of scaffolds with or without UV treatment to soak up liquid was evaluated using ddH_2_O, hypertonic solution, and blood. Liquids were placed into 35 mm petri dishes on a flat surface, and the undersurface of the scaffold was placed on the liquid surface and kept in contact for one minute. Then, the scaffold weight, including the liquid, was subtracted from the original weight to measure liquid absorption into the scaffold per scaffold volume; 70% glycerol (Wako, Japan) was used as a hypertonic liquid, and blood was obtained from rats.

### 2.2. Osteoblast Cell Culture

Primary osteoblasts were cultured from rat bone marrow stromal stem cells as previously reported [55,110,111,112,113]. Femurs were excised aseptically from 8-week-old male Sprague Dawley rats. Soft tissues and muscles were removed and washed with 1× phosphate buffered saline (PBS; Gibco, Carlsbad, CA, USA) several times. Both ends of the femur bone were carefully severed, and the bone marrow was flushed out using 3 mL of osteoblastic differentiating medium through a 20-gauge needle. The osteoblastic differentiating medium contained a modified Eagle’s medium, 15% fetal bovine serum (FBS), 50 μg/mL ascorbic acid, 10^−8^ M dexamethasone, 10 mM Na-β-glycerophosphate, and antibiotic–antimycotic solution containing 10,000 units/mL penicillin G sodium, 10,000 mg/mL streptomycin sulfate, and 25 mg/mL amphotericin B. The extracted cells were cultured in 100 mm dishes, and at 80% confluency, the cells were detached using 0.25% trypsin 1 mM EDTA 4Na and passaged. After 2-time passaging, the cells were finally seeded onto Ti microfiber scaffold disks (20 mm diameter, 1.5 mm thick) and placed in a 12-well culture dish at a density of 3 × 10^4^ cells/cm^2^ for WST-1 and ALP assays. Cells suspended in 1 mL of culture medium per well were poured along the wall of the well. The culture medium was renewed every 3 days. Untreated and UV-treated titanium scaffolds were compared.

### 2.3. Number of Attached and Propagated Cells, Cell Attachment Assay and Osteoblast Behavior on Titanium Microfiber Scaffolds

The number of osteoblasts present on the scaffolds was evaluated after 3 h and 24 h of culture using a WST-1-based colorimetric assay (WST-1, Roche Applied Science, Mannheim, Germany). WST-1 assay is based on the metabolic activity of cells and used for a wide range of quantification that includes cell viability, attachment, and proliferation, based on the assumption that the number of cells is correlated with the amount of their metabolic activity [54,114,115,116,117]. Scaffolds in tissue culture wells were incubated at 37 °C for 1 h with 100 μL of WST-1 reagent. The amount of formazan product was measured with a multi-detection microplate reader (Synergy^TM^ HT, BioTek Instruments, Inc., Winooski, VT, USA) at a wavelength of 450 nm. Similarly, the number of propagated cells on day 2 of culture was measured using WST-1 assay.

Osteoblasts on titanium microfibers were also observed by fluorescence microscopy on day 2. Cells were fixed in 10% formalin and stained with the fluorescent dye rhodamine phalloidin (actin filament, red color; Molecular Probes, Eugene, OR) and 4′,6-diamidino-2-phenylindole (DAPI) (nuclear, blue color; Abcam, Cambridge, UK). Fixing with formalin was proven effective, particularly for immunostaining combined with fluorescent dye in our previous studies [52,118,119].

### 2.4. ALP (Alkaline Phosphatase) Activity

ALP activity on day 4 was examined. Cultures were rinsed with ddH_2_O and 250 mL of p-nitrophenylphosphate was added; then, the samples were incubated at 37 °C for 15 min. ALP activity was calculated as the quantity of nitrophenol measured at a wavelength of 405 nm using a microplate reader.

### 2.5. Real-time Quantitative Polymerase Chain Reaction (qPCR)

Gene expression was analyzed by qPCR on day 7. Total RNA was extracted from cells using TRIzol reagent (Invitrogen, Carlsbad, CA, USA) and the Direct-zol RNA MiniPrep kit (Zymo Research, Irvine, CA, USA). Extracted RNA was reverse transcribed into first-strand cDNA using SuperScript III Reverse Transcriptase (Invitrogen). The quantitative PCR reaction was performed in a 20 μL volume containing 90 ng cDNA, 10 μL TaqMan Universal Master Mix II, and 1 μL TaqMan Gene Expression Assay using the QuantStudio 3 Real-Time PCR System (Thermo Fisher Scientific, Canoga Park, CA) to quantify osteopontin (*Spp1*), osteocalcin (*Bglap*), and *Runx2* mRNA expression. *Gapdh* expression was used as the endogenous control.

### 2.6. Animal Experiments

Male Sprague–Dawley rats, 12 weeks old, were anesthetized by inhalation of 2–3% isoflurane. Their right legs were shaved and scrubbed with 10 % povidone–iodine. The distal aspects of femurs were exposed via a skin incision and periosteum separation. For the purpose of a vertical bone augmentation model, a plate form of titanium microfiber scaffolds (5 × 3 × 1.5 mm) was fit and placed on a flat surface of the femur (approximately 77–12 mm from the knee joint), avoiding the distal epiphysis, and fixed with a Ti6Al4V screw (1.5 mm diameter, 5 mm length) (KLS Martin, Tutllingen, Germany) with bi-cortical support (Figure 1a). Blood and cells could only be supplied through the screw and unidirectionally. After 2 weeks of healing, the femurs were removed. After the screw was carefully removed from the Ti scaffold, the mechanical strength of osseointegration between bone and scaffold was examined with a testing machine (Instron 5544 Electro-mechanical Testing System, Instron, Canton, MA, USA) equipped with a 2000 N load cell. A 0.8 mm diameter pushing rod was used to load the scaffold at 1 mm away from the screw vertically downward at a crosshead speed of 1 mm/min (Figure 1b,c). The “broken” value between the bone and the scaffold surface was determined by measuring the peak of the load-displacement curve. Additionally, after carbon sputter-coating, the contact aspects of the bone and scaffolds were examined by scanning electron microscopy (SEM) (XL30, Philips, Eindhoven, Netherlands), and the elemental composition of the scaffold and connected tissue were analyzed by energy-dispersive x-ray spectroscopy (EDX).

### 2.7. Statistical Analysis

Culture studies were performed in triplicate (*n* = 3) for each group (untreated vs. UV-treated). Four animals in each group were tested in animal experiments (untreated vs. UV-treated). Differences between untreated and UV-treated groups were compared by one-way ANOVA after confirming the normal distribution of each group. *p*-values < 0.05 were considered significant.

## 3. Results

### 3.1. Morphology, Surface Chemistry, Wettability, and Fluid Absorption of Titanium Microfiber Scaffolds

Low-magnification SEM images of the scaffolds showed even, dense networks of titanium fibers (Figure 2a). High-magnification images showed that the titanium fibers were of consistent ~20 µm diameter/thickness (Figure 2b). The pore size (space between fibers) was 20–50 µm.

XPS revealed significant differences in the carbon peak between untreated and UV-treated microfibers (Figure 3). There was a high intensity of carbon (C1s) on untreated Ti microfiber, higher than oxygen (O1s) and titanium (Ti2p) (Figure 3a); ~30, 40, and 20% carbon, oxygen, and titanium on untreated Ti microfibers. UV treatment decreased the amount of carbon and increased the amount of oxygen and titanium to ~10, 50, and 30%, respectively (Figure 3b,c).

Water droplets remained on the top of untreated scaffolds and were not absorbed into the scaffolds (Figure 4a), while water droplets were completely absorbed into UV-treated scaffolds (Figure 4b). The contact angle between the droplet and untreated scaffold was approximately 120° compared with 0° on UV-treated scaffold (Figure 4c). Fluid absorption of water, glycerol, and blood increased significantly (by 5–7 times) into titanium microfiber scaffolds after UV treatment (Figure 5a–c).

### 3.2. Osteoblast Attachment and Propagation

The number of osteoblasts attaching to titanium microfiber scaffolds was measured 3 and 24 h after cell seeding. There were more cells on UV-treated microfibers than untreated ones (Figure 6a). On day 2, the number of propagated osteoblasts was also significantly greater on UV-treated microfibers (Figure 6b). Fluorescence microscopy on day 2 showed that more osteoblasts were present on UV-treated titanium microfibers than on untreated ones (Figure 6c,d).

### 3.3. Osteoblast Differentiation

ALP activity was significantly higher in osteoblasts grown on UV-treated microfibers than untreated microfibers (Figure 7a). Osteocalcin gene expression was significantly greater in osteoblasts on UV-treated Ti than untreated Ti (Figure 7c), although there were no significant differences in osteopontin and Runx2 expression (Figure 7b,d).

### 3.4. Mechanical Strength of Osseointegrated Scaffold and Interfacial Tissue Morphology

Load-displacement curves from shear testing at week 2 of healing (Figure 8a,b) revealed that the mean strength of the microfiber scaffold–bone interface for UV-treated microfibers was approximately twice that of untreated ones (Figure 8c).

Regarding the result, let us hypothesize that the scaffold-to-bone interfacial strength was enhanced because of the de novo bone ingrowth into the scaffold integrating the innate cortical bone surface and scaffold. Therefore, the interfacial tissue morphology and chemistry between the innate femoral cortical bone and microfiber scaffolds were observed by SEM and EDX, respectively (see representative images, Figure 9). SEM images showed that most of the individual microfibers in the untreated group were identifiable and were not covered with de novo tissue (Figure 9a). Biological tissues seamlessly occupied the spaces between microfibers in the UV-treated group, with pre-existing pores extensively filled (Figure 9b). We also noted the presence or absence of a gap between the innate cortical bone and scaffolds: there was a clear, large gap along the entire untreated scaffold interface, whereas the gap was narrower and almost absent at the UV-treated scaffold interface (Figure 9a,b).

EDX mapping for elemental Ti showed that titanium microfibers were more masked in the UV-treated group than the untreated group (Figure 9c,d). Magnified SEM images confirmed the presence of large gaps between the untreated scaffold and innate bone (Figure 9e), whereas new bone infiltrated the microfiber network in the UV-treated group (Figure 9f).

## 4. Discussion

Here, we report that UV treatment enhances osteoblast behavior and osteoconductivity of titanium microfibers and their scaffolds in a vertical augmentation model with very limited blood and cellular supply from around a screw. Titanium surfaces lose osteoconductivity over time due to the progressive accumulation of hydrocarbons on their surfaces, which is also known as “biological aging” of titanium [62,94,97,98,116]. This accumulation of hydrocarbons can be cleaned by UV treatment [10,82,96,120,121,122]. Here we confirm that UV light can also decarbonize a 3D network of titanium microfibers and rejuvenate superhydrophilicity of the microfibers. Even a highly hypertonic liquid infiltrated into UV-treated, superhydrophilic scaffolds, while untreated, hydrophobic Ti microfiber scaffolds repelled the liquids and prevented absorption. Culture medium easily infiltrated into UV-treated scaffolds, permitting osteoblasts to be attracted and attach to the microfibers. Osteoblasts attach on flat titanium surfaces within 24 h [13,101,102,123], where they spread, proliferate, and develop a mature cytoskeleton. This study demonstrated that osteoblasts can settle and proliferate even on the thin, 20 mm microfibers with an aid of UV treatment. Given that osteoblasts commonly spread as wide as 50-100 mm on culture dishes or biomaterials, the successful and rapid attachment of osteoblasts with robust osteogenic function on 20 mm fibers is highly significant. It was even more notable that cells were detected on the scaffolds even 3 h after seeding. Our previous studies demonstrated a 3 h culture was sufficient for the attachment and initiation of the spread of osteoblasts on UV-treated titanium disks and was validated as an effective time point to show a contrast between untreated and UV-treated groups [74,76,78,101]. Based on the experience, we attempted a challenging time point of 3 h in addition to 24 h. Considering the WST-1 values on day 2 ranging from 0.05 to 0.09 (Figure 6b) which was accompanied with the clear visualization of cells by fluorescent microscope, we believe that WST-1 values ranging from 0.01 to 0.04 at 3 and 24 h are valid to indicate the presence of cells on the scaffolds. UV treatment converts the electrostatic Ti surface from a negative to a positive charge and probably promotes helping to recruit and retain cells [70,74].

It has been suggested that rapid osteoblast attachment accelerates the development of their activity and mineralization [87,121]. We found that the ALP activity of osteoblasts in UV-treated scaffolds was double that of cells in untreated scaffolds. Furthermore, the expression of osteocalcin—a late-stage differentiation marker associated with cell cycle arrest in osteoblasts [124,125]— was significantly higher in cells in UV-treated scaffolds. Because there is an inverted relationship between osteoblastic differentiation and proliferation, the more proliferation, the slower the differentiation [17]. Generally, osteoblasts tend to start functioning when they stop proliferating, although both proliferation and mineralization need to be balanced for effective bone engineering. Indeed, Hirota et al. [126] reported that hydroxyapatite (HA)-coated thin Ti fibers accelerated osteoblast differentiation, whereas osteoblasts attached to untreated thin Ti fibers proliferated more. This study consistently showed that UV treatment increased the number of osteoblasts on the scaffolds, and therefore, we postulate that osteoblasts grew more rapidly to form colonies and clusters on UV-treated scaffolds, increasing the cell-to-cell interaction, and as a secondary consequence, resulted in accelerated differentiation. Potential effects of increased cell-to-cell interaction on UV-treated microfibers on the osteoblastic differentiation is of interest in future studies.

The expression of osteocalcin and Runx2 was not modulated by UV treatment. Osteopontin, a mid-stage marker of osteogenic differentiation, shows a two-phasic pattern of expression with an early and late upregulation and a dip in between. This study examined the gene expression at only the time point of day 7, and a further study with an analysis at multiple time points is necessary. Runx2 is an essential transcription factor for osteoblastic differentiation. Runx2 promotes the early differentiation from mesenchymal stem cells to osteogenic progenitor cells, and the continuous expression of Runx2 maintains the osteogenic progenitor cells in their immature stage [127,128,129]. Afterward, along with the downregulation of Runx2, osteoblasts differentiate to more mature stages, and Runx2 is no longer necessary [129]. As shown in the robust osteocalcin expression, osteoblasts examined in the present study were assumed to be in the mature stage, when the role of Runx2 was nearly complete, resulting in the un-modulated expression by UV treatment. Again, a study to examine an earlier stage gene expression will provide us with more information. Given that osteoblasts are not expected to proliferate effectively on thin microfibers due to the limited surface area, the number of initially attaching cells may determine the quantity and quality of final matrix mineralization. In this context, the carbon-free, superhydrophilic status of the scaffold obtained by the UV treatment that recruited more cells was a significant advantage.

Surface topography of titanium also influences osteoblast function. For example, micro-scale roughness produced by acid etching accelerates osteoblast differentiation and mineralization compared to relatively smoother titanium surfaces, such as a machine-milled surface [12,17]. In addition, the hardness and stiffness of mineralized tissue formed on micro-roughened titanium surfaces are increased compared to those on machined surfaces [8,12,17]. Therefore, we considered roughening microfiber surfaces in the present study. However, acid etching of titanium fibers was expected to reduce the fiber diameter, which significantly reduces the surface area available for osteoblasts to inhabit. We also found that subtractive roughening, such as acid-etching, on 20 mm thin titanium microfibers was technically near impossible. Therefore, we used the microfiber scaffolds as made without surface roughening as shown in the SEM images (Figure 2b). The present study demonstrated that osteoblasts were well differentiated even on the smooth-surfaced microfiber scaffolds, as shown in the osteocalcin expression, with an aid of UV treatment. More importantly, as we mentioned in the Introduction, 3D architecture made of fibers and pores may have served a micro-rough configuration mimicking the acid-etch-created microrough surface.

The osseointegration of the UV-treated scaffold was significantly greater than that of the untreated scaffold. A clear gap was observed between untreated scaffold and bone tissue, and the mechanical strength of the connection between the bone and the untreated scaffold (~4 Ncm) was less than half of that of the UV-treated scaffold (~10 Ncm), indicating that the untreated scaffold rarely connected to the bone, but there was new bone formation in the UV-treated scaffold due to osseo-infiltration. In the present model, osteoblast recruitment may have been hampered (1) because the scaffold may not be rigidly immobilized due to a single screw used to fix the scaffold, and (2) blood and cells supply was unidirectional and could only gain access to the scaffold from around the screw (Figure 1a). Even under this challenging condition, the bone tissue infiltrated into the UV-treated scaffold, suggesting a synergy effect between the titanium microfiber scaffold and the UV photofunctionalization on vertical bone augmentation.

Regarding the experimental design, we considered including a well without a titanium scaffold as a control. However, the scaffold disk (20 mm in diameter) used in this study was smaller than a well (22 mm in diameter) of 12-well culture plates, which makes the comparison difficult. More importantly, in our preliminary study, remarkably fewer cells attached to the scaffold than to a culture dish. When the assay protocols, such as WST-1 and ALP detection, were optimized to the culture dish control, the experimental groups of untreated and UV-treated scaffolds remained below the detection limit. Therefore, considering the sensitivity and validity of the method, we focused on the comparison between the untreated and UV-treated scaffolds.

Osteoconductivity has been shown to be improved by surface curvature [55,56,130], and laser-treated gyroid Ti scaffolds enhance osteoconduction [131], which may be an advantage of using the fibered titanium instead of flat titanium. Pore size or porosity are likely to affect bone formation in the scaffold [132,133]. The present scaffold had 87% porosity, which produced favorable osteoblastic behavior [134]. Further in vivo investigations of bone ingrowth into UV-treated thin microfiber scaffolds, including histological assessment, are needed to explore optimal microfiber scaffold configurations and applications. Clinical application of bone engineering remains in its infancy. Based on the present results, future studies on the application of UV-treated titanium microfiber scaffolds which have more complicated 3D structures will be planned.

## 5. Conclusions

UV treatment of Ti microfiber scaffolds enhanced osteoblast recruitment and attachment, which allowed osteoblasts to proliferate and express genes related to bone formation within the scaffold. UV-treated microfiber scaffolds promoted ALP activity and rapid osteocalcin expression compared with untreated scaffolds. In a vertical augmentation model in rats, osteoconductivity was enhanced by UV treatment, and the biomechanical strength of integration between the innate cortical bone and regenerated bone inside the scaffold was twice that of bone forming in untreated scaffolds. This study established proof-of-concept of the synergy between titanium microfiber scaffolds and UV photofunctionalization to vertically augment bone, and further studies are now warranted to optimize the scaffold and UV treatment conditions for better outcomes.

## Figures and Tables

**Figure 1 cells-12-00019-f001:**
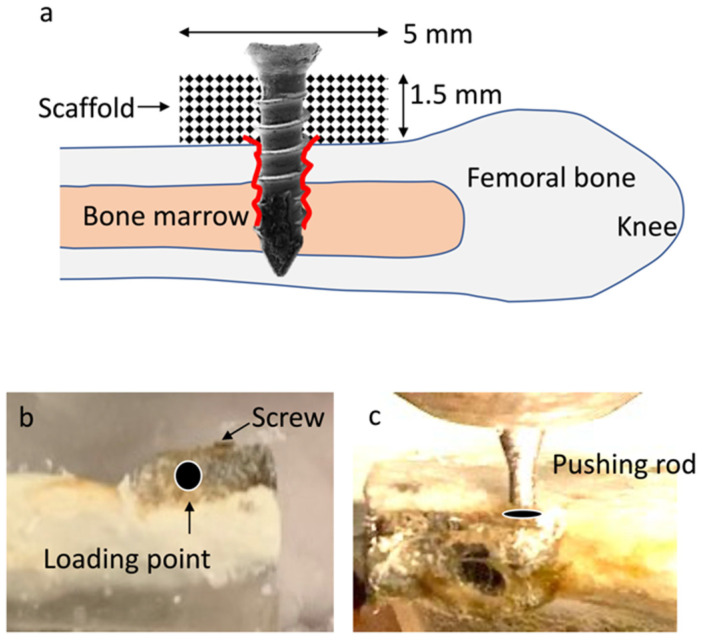
(**a**) Schema of the animal experiments simulating a vertical bone augmentation. A plate-form scaffold was fit and placed. The scaffold (arrow) was placed on a flat surface of innate cortical bone and fixed with a titanium screw. Note blood and cells could only gain access to the scaffold through the screw in one direction; (**b**) a photograph of the retrieved femur with an integrated titanium microfiber scaffold; (**c**) after the screw was removed, the mechanical strength between the cortical bone and scaffold was measured by vertically pushing down the scaffold (at the indicated loading point 1 mm away from the screw).

**Figure 2 cells-12-00019-f002:**
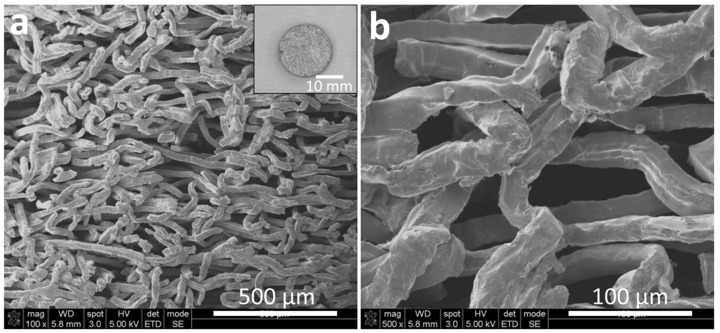
SEM images of the titanium microfiber scaffold; low-(**a**) and high-(**b**) magnification images of angular titanium microfibers forming a non-woven, fabric-like structure. A photograph of a disk-form scaffold is also shown (**a**).

**Figure 3 cells-12-00019-f003:**
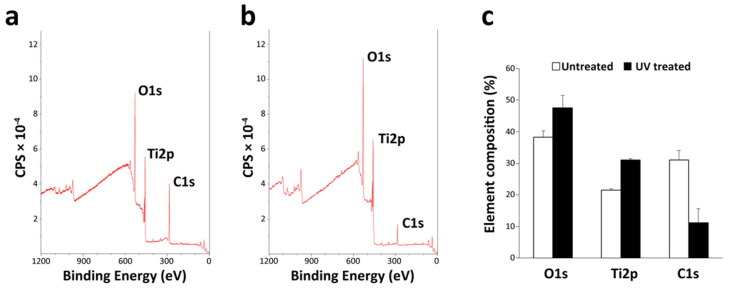
XPS analysis of titanium microfiber scaffolds. XPS spectra of untreated (**a**) and UV-treated (**b**) scaffolds. Carbon is prominent on untreated scaffolds compared with UV-treated scaffolds; (**c**) elemental composition by XPS analysis of untreated and UV-treated scaffolds. Oxygen (O1s) and titanium (Ti2p) are increased on UV-treated scaffolds, whereas carbon (C1s) is reduced on the UV-treated scaffolds, indicating UV treatment removed carbon on titanium microfiber surfaces.

**Figure 4 cells-12-00019-f004:**
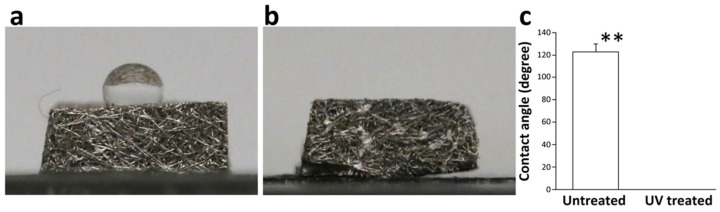
Wettability of the scaffold; (**a**) a water droplet forms on the untreated scaffold due to its hydrophobicity; (**b**) no water droplet is seen on the UV-treated scaffold due to its superhydrophilicity; (**c**) contact angle between the water droplet and the untreated scaffold was approximately 120°, whereas it was 0° on the UV-treated scaffold because the UV-treated scaffold completely absorbed the water droplet. ** *p* < 0.01.

**Figure 5 cells-12-00019-f005:**
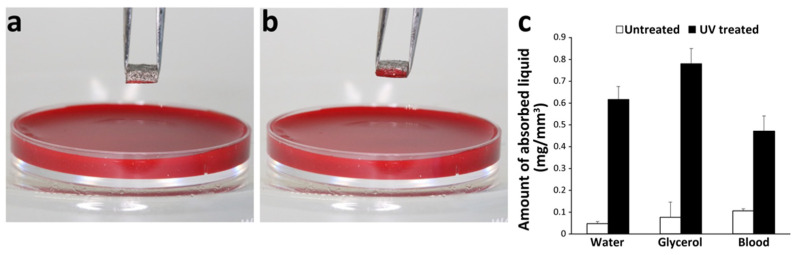
Liquid recruiting capability of the scaffold. Photographic images of blood absorption into untreated (**a**) and UV-treated (**b**) scaffolds. Blood adhered to the undersurface of the untreated scaffold in contact for 1 s, while the blood was absorbed halfway inside the UV-treated scaffold, indicating that UV treatment enhanced blood recruitment; (**c**) amount of water, glycerol, and blood absorbed into untreated and UV-treated scaffolds. All liquids were absorbed considerably more into the UV-treated scaffolds.

**Figure 6 cells-12-00019-f006:**
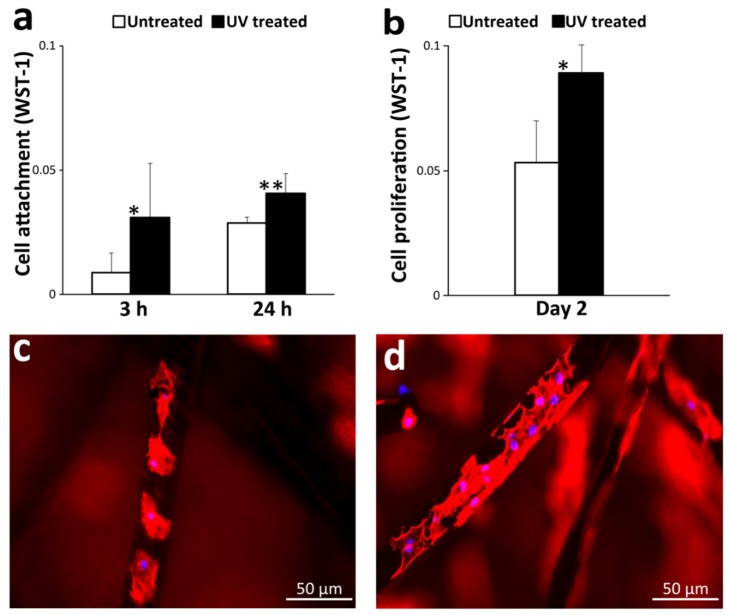
Cell attachment *and propagation* of rat-bone-marrow-derived osteoblasts on untreated and/UV-treated Ti microfiber scaffolds; (**a**) the number of cells attached to each scaffold with or without UV treatment at 3 and 24 h was evaluated by the WST-1 assay; (**b**) the number of osteoblasts propagated on scaffolds on day. (**c**,**d**) visualization of cells on day 2 using fluorescent microscopy, showing more cells attached to UV-treated titanium microfibers (**d**) than to untreated ones (**c**); * *p* < 0.05, ** *p* < 0.01.

**Figure 7 cells-12-00019-f007:**
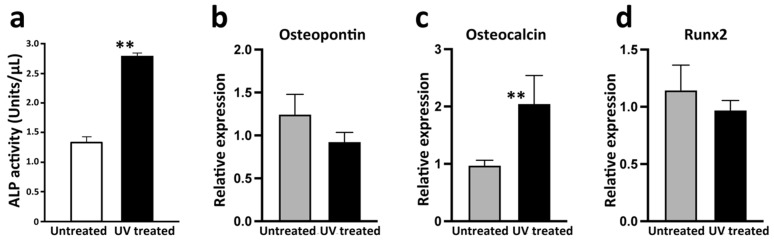
Osteoblastic differentiation evaluated by ALP activity and gene expression analysis by qPCR. ALP activity of osteoblasts in the UV-treated scaffold was significantly greater than that in the untreated scaffold (**a**). There was no difference in gene expression of osteopontin and Runx2 between the two groups (**b**,**d**), while expression of osteocalcin in the UV-treated scaffold was significantly greater than that in the untreated scaffold (**c**). ** *p* < 0.01.

**Figure 8 cells-12-00019-f008:**
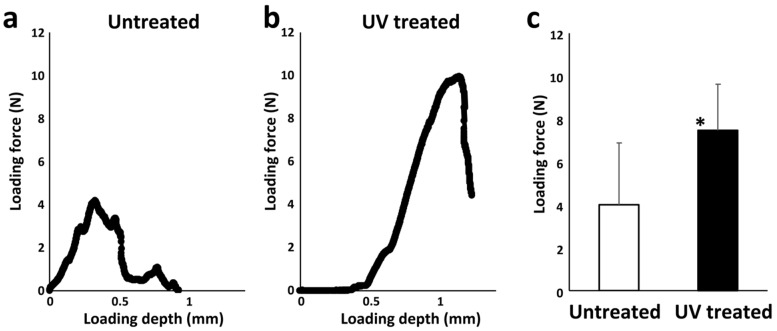
Biomechanical strength of the connection between scaffold and bone in a vertical bone augmentation model. Representative load-displacement curves are shown (**a**,**b**). The average loading force to break the titanium microfiber scaffold from the surface of cortical bone was approximately 4 N for untreated scaffold and 8 N for UV-treated scaffold (**c**); * *p* < 0.05.

**Figure 9 cells-12-00019-f009:**
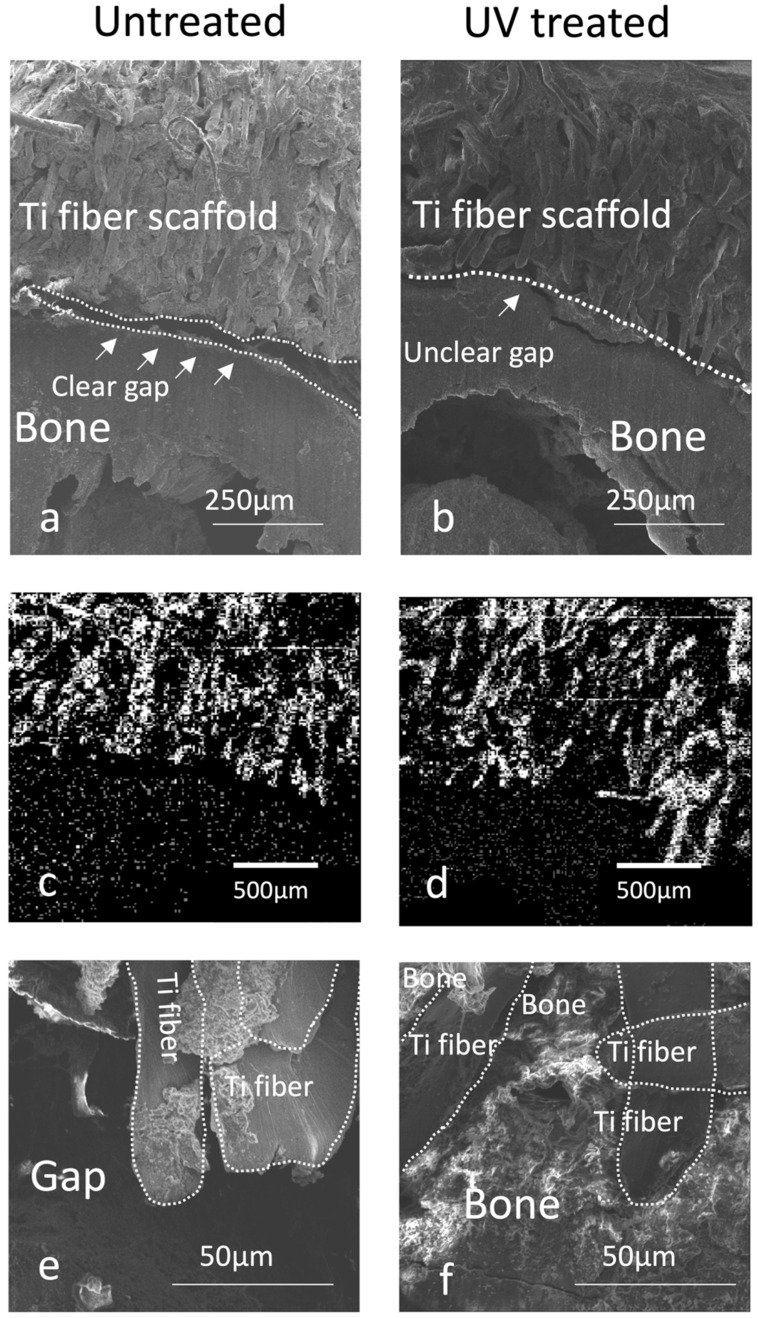
Morphological and chemical analysis of the titanium microfiber scaffold and cortical bone interface after a 2-week healing in a vertical bone augmentation model. A clear gap is visible between the bone and untreated scaffold (**a**), while the gap between the bone and the UV treated scaffold is unclear (**b**). EDX of Ti showed a relatively clear border between the bone and titanium fibers (**c**) compared with the UV-treated scaffold (**d**), in which titanium fibers apparently infiltrate into the bone. The bone tissue is sparsely formed on the untreated Ti microfibers with no connection to the bone (**e**), while UV-treated Ti microfibers apparently connect to the bone tissue (**f**).

## Data Availability

Not applicable.

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
