# Peer review of "Ultraviolet Light Treatment of Titanium Microfiber Scaffolds Enhances Osteoblast Recruitment and Osteoconductivity in a Vertical Bone Augmentation Model: 3D UV Photofunctionalization"

_cells, 2022, doi:10.3390/cells12010019_

Round 1

Reviewer 1 Report

The objective of the study was to evaluate the capacity 18 of a UV-photofunctionalized titanium microfiber scaffold to recruit osteoblasts, generate intra-scaffold bone, and integrate with host bone in a vertical augmentation model with unidirectional, limited blood supply.

Altough the objective sounds interesting, methodological approach lacks consistence and a number of items, both in the material and in the biological parts , should be fixed prior to paper admittance for publication, if possible .

Please find listed below some of the sugerences and questions :

 Methods :

Section 2.1. Titanium characterization is limited to SEM study of surface. It is not clear how many experimental groups are employed

Section 2.3. Clarify cell isolation methodology, cell seeding , experimental groups, control groups and timing

Section 2.5 . Cell attachment appears to be quantified using WST -1 after 3 and 24 h and maybe 2 days (not in graph ) . WST is a well known cell viability marker, but not an adhesion marker. Maybe the authors could explain the decision. Why formalin and not methanol for fixing ?

Results

 Section 3.1

Figure 3 : discordance in y axis scale.Please unify parameters as graphics in the present form are confusing as the Ti peak appears quite similar in heigth in both graphs

Section 3.2.

Overall the whole section appears to be speculative.

No control groups without Ti fibers is presented , please indicate the reason . A control group should be incorporated for quality control of the methodology employed .

Cell attachment appears to be quantified using WST -1 after 3 and 24 h . WST is a well known cell viability marker, but not an adhesion marker. Maybe the authors could explain the decision .

In the methods section no clear description of cell seeding is present. In usual protocols a drop of cell suspension is seeded onto scaffolds placed into plates and, after an interval not less than 1 h, cell media is completed. If this was the case for the experiments presented, it is not clear that after 3 hours after seeding an efficient cell response could be achieved. Please clarify and incorporate Raw data for figure 6a , together with data obtained from at least a control group/time

 No data for cell attachment on day 2 are presented , but rhodamine labelled cells are analyzed at day 2 . Please clarify this point

 Figure 6: This is not a microscopic image ( line 253) , as we can apreciate the image with naked eye. Maybe the author refer to a photomicrography?

 The authors refer to cell attachment: “A few cells attached without adequate spreading on untreated Ti micro- 254 fibers (c), whereas cells were well spread and densely attached”

In this sense:

No markers of cell attachment are presented. How can the authors identify “ densely attached “ cells ? In the figure, cell morphology appears to be quite similar in both images , no markers for cell adhesion, no cytoskeletal changes but apparently a higher number of cells . A quantitative approach , maybe nuclear count, cell surface or surface area covered by cells , etc… could help to clarify this point

What can we consider an “adequate spreading “? Is there any control or experimental group with inadequate spreading ?

Section 3.3.

Figure 7: no differences in Runx and osteopontin levels . Could the authors include an explanation in the results or discusion section ?

Section 3.4

Figure 8. Please confirm scale for 8a and 8b

 Discussion

Line 315: no attachment is demonstrated in the study

Line 320: “Uv treatment …..probably helping to promote a recruitment and retention of cells“ . Please consider that this point should be asserted and non speculative

Line 333

“This study consistently showed that UV treatment increased the quantify of osteoblasts” What do they mean with “quantify?

Line 335

“increasing the cell-to-cell interaction, and as a secondary consequence, resulted in the accelerated differentiation”As indicated above no cell to cell interaction is demonstrated in the draft

Paragraph between lines 335 and 340 should be explained and clarified. Also paragraph 341-353. They are contradictory both with background in the field regarding acid etched or rough surfaces and also with the results presented in previous sections. Please rewrite and clarify

Line 340 : The authors say: “In this context, the 338 carbon-free, superhydrophilic status of the scaffold obtained by UV treatment effectively 339 increased the bioactivity of the microfiber scaffolds.”

No bioactivity assays (OH apatite nucleation in SBF or similar …) are described in the text. In the case that the authors have unpresented data is this aspect, please show

Line 349 and following  : “Our fluorescence microscopy after 2 days of culture revealed that osteoblasts  fully and densely attached or proliferated in UV-treated scaffolds, suggesting that the smooth surfaces of the microfibers used in this study were sufficient to promote osteoblast  bioactivity”.

Osteoblasts do not have bioactivity . That is a material inherent, or induced, feature .

Do they attach or proliferate ? or both? Please clarify

Line 359: “Nevertheless, bone formation in the scaffold was poor even for UV-treated scaffolds”.

How does it matches with lines 362-365? Please clarify

Finally , in line 369 , the authors indicate : “The present scaffold had 87% porosity, which produced favorable osteoblastic behavior”

Where does this data comes from? Please indicate and show the results for Textural characterization performed by nitrogen physisorption or similar methodology for porosity determination

Final recomendation : major changes

Reviewer 2 Report

Ultraviolet light treatment of titanium microfiber scaffolds enhances osteoblast recruitment and osteoconductivity in a vertical bone augmentation model: 3D UV photofunctionalization

The manuscript seems to be interesting. It describes the effect of UV treatment in improving the proliferation and differentiation of osteoblasts in Ti microfiber scaffolds.  The current manuscripts can be published in the journal of Cells once a major revision is done. The reviewer’s comments are listed as follows:

General comment

1-      As the authors also performed the capability of the 3D model in in-vivo, might be better to present a figure showing this.

2-      To compare the proliferation of the cells on the scaffold (in the abstract), might be better to mention the result of longer time points such as 2 days (instead of 3h)

3-      To sterilize samples, ethanol and or UV light are usually used. The reviewer is curious which sterilisation protocol has been used here?

4-      The manuscript needs a proof-read before publishing  such as a lack of spacing before referencing and so forth).

Abstract

·       The tense of the verbs in Abstract better to be changed (e.g., the objective of this study was…)

·       Scaffolds were fabricated from 20 mm grade 1 commercially pure titanium.. could be rephrased.

·       As details of the experiments in the abstract have been provided, then expected to bring the information about the uv source (e.g., wavelength and laser power)

·       Which fabrication technique did the authors use for the scaffolds?

Keywords:

Some of the keywords have not been mentioned in the abstract. Seems like irrelevant to the content of the study (e.g, dental and orthopedic implants).

Materials and method

1-      Why did the authors not use same disc for both in-vitro and in-vivo tests?

2-      Might be better to shortly explain how the scaffolds were fabricated.

3-      Just as a curiosity, it seems the authors pre-cultured the cells in osteogenic medium. Is it a standard for such kind of cells?

Results

1-      In Figure 2, it might be also interesting if bring another image showing the entire scaffold (higher magnification). Moreover, the SEM images might be edited.

2-      A remarkable difference is shown in Figure 8, for the force-displacement. Can the authors interpret the results? Why does UV exposure increase mechanical strength?

3-      It seems it can be better if the authors bring the mechanical test results before the biological results (for the sake of discussion). 

Reviewer 3 Report

I was pleased to review the article ID cells-2119218 entitled “Ultraviolet light treatment of titanium microfiber scaffolds enhances osteoblast recruitment and osteoconductivity in a vertical bone augmentation model: 3D UV photofunctionalization” for the Cells Journal. The article investigates UV light as a treatment for titanium microfiber scaffolds to improve vertical bone formation.

Overall, the article is well organized, with a thorough introduction and adequate methodology.

Please include the unit in the Y axis in Figures 6, and 7.

If possible, the authors could add histological Hematoxylin and Eosin or Masson’s Trichome stained slices for better in vivo results interpretation.

Round 2

Reviewer 1 Report

The authors have re-organized the Materials and methods section according to suggestions. Now the section is clearly written and experimental design appears more clear. 

The results sections has been also rewritten and modified according to suggestions, and mainly section 3.2 is clarified . Figures are now clear and according to text 

Discussion section has been modified and clarified according to suggestions. 

The paper is now acceptable for publication awaiting for editor decision